# Evaluation of the Femoral and Tibial Alignments in Dogs: A Systematic Review

**DOI:** 10.3390/ani11061804

**Published:** 2021-06-17

**Authors:** Masoud Aghapour, Barbara Bockstahler, Britta Vidoni

**Affiliations:** 1Small Animal Surgery, Department for Companion Animals and Horses, University of Veterinary Medicine, 1210 Vienna, Austria; Britta.Vidoni@vetmeduni.ac.at; 2Section of Physical Therapy, Small Animal Surgery, Department for Companion Animals and Horses, University of Veterinary Medicine, 1210 Vienna, Austria; Barbara.Bockstahler@vetmeduni.ac.at

**Keywords:** femoral angles, tibial angles, limb alignment, bone deformity, dog

## Abstract

**Simple Summary:**

The measurement of limb alignments is an important topic in veterinary orthopedics. These measurements enable veterinarians to assess normal limb functions, diagnose congenital or acquired disorders, and plan proper treatment protocols. Different measurement methods have been reported for fore- and hindlimb measurements in the literature, and reference ranges have been published for different breeds. These standard values can be compared with the measured values in small animal clinics, especially in the case of bilateral deformities, in which a sound extremity does not exist to provide a reference value. In this review, we aimed to compile the relevant values from the literature, sorting them according to the dog breed and the health status of the dog.

**Abstract:**

The assessment of limb conformations in veterinary orthopedics is a significant tool used to determine the quantitative degree of limb malalignments. As in human medicine, various studies have been undertaken in veterinary medicine to determine the values in different dog breeds and to determine the values in healthy and diseased dogs. The objectives of this systematic review were to evaluate the reported values in these articles separately, to compile the standard values, and to compare the values between dogs with and without various orthopedic diseases. All of the articles included in this systematic review were collected by screening the Scopus, PubMed/Medline, and Web of Science databases. The articles were evaluated according to the measured alignments, imaging methods, dog breeds, and the health status of the dogs. Each alignment was investigated separately, and the results are summarized. Twenty-nine studies were included in this systematic review. According to the studies, in the frontal plane, distal femoral alignments, as well as proximal and distal tibial alignments, corresponded to the severity of the medial patellar luxation. The difference between affected and non-affected dogs with cranial cruciate ligament disease was limited to the proximal tibial alignments in the sagittal plane.

## 1. Introduction

The evaluation of fore- and hindlimb conformations and clinical goniometry have long been important topics in veterinary orthopedics. Having reference values of pelvic limb alignments, including anatomical and mechanical angles of the femur and tibia, would help veterinarians to specify the quantitative degree of malalignments. In human medicine, different methods have been developed to quantify the degree of upper and lower limb deformities [1]. Using standard measurement methods provides reliable and homogenous values for surgeons and allows them to use these reported scales in clinics, especially in the case of bilateral deformities, in which a sound extremity does not exist to provide a reference value. Normal limb alignments may vary in different dog breeds or between large and small breeds; therefore, the determination of reference ranges for different dog breeds is important.

Angular deformities of the canine hindlimb have mostly been reported in the femur, tibia, and metatarsus [2,3,4,5,6,7,8,9,10]. These deformities usually develop after premature total or partial closure of the physis. Physeal damage resulting from premature closure can occur due to various causes, including trauma, nutritional imbalances, hypertrophic osteodystrophy (metaphyseal osteopathy), retained cartilage cores, and iatrogenic reasons, such as an improper application of a fixation apparatus [11,12,13]. In addition, uneven tension and pressure conditions that act on the distal femoral physis during growth can lead to angular deformity of the femur [14]. Distal femoral varus is defined as the inward angulation of the distal femur toward the body. Abnormal distal femoral varus may be associated with medial patellar luxation (MPL) [14]. Angular deformity of the femur in the frontal plane (varus/valgus) can be diagnosed based on joint reference angles, including the anatomical and mechanical proximal or distal femoral angles. The joint reference angle is an angle between the bone axis and its respective joint orientation lines [1]. The bone axis may be the mechanical or anatomical axis. The mechanical axis is a straight line connecting the centers of the proximal and distal joints of the bone. The anatomical axis is a straight or curved line that passes through the center of the bone [1]. Bone deformities in the canine pelvic limb are not limited to the femur. Pes varus and valgus describe skeletal deformities, characterized by a medial and lateral deviation of the bone axis of the distal tibia in the frontal plane, respectively [7,8,9]. The etiology of this skeletal deformity is an asymmetric growth of the distal tibial physis because of traumatic, nutritional, or developmental premature closure of the physis [4,5,8].

Various studies have been carried out to evaluate the femoral and tibial alignments in dogs. In general, these studies have focused on the development of measurement methods, investigating the difference between sound and diseased dogs, and the accuracy of different methods or tools. The current study was carried out according to the PRISMA guidelines [15] and focused on the previously performed studies involving standardized methods of measurement and terminology relating to pelvic limb alignments in dogs. In this review, we aimed to evaluate each alignment that has been reported in the articles separately, to report standard values in healthy dogs, and to compare the measured values in dogs with and without different orthopedic diseases.

## 2. Materials and Methods

The standard guidelines for reporting systematic reviews and meta-analyses (PRISMA Statement), reported by Moher et al. [15], were followed in this study.

### 2.1. Data Sources

All articles were collected by screening the Scopus, PubMed/Medline, and Web of Science databases on 24 September 2018.

### 2.2. Search Strategy

A Scopus search for the term ‘alignment or malalignment or angle or angular value’ yielded 2,625,647 articles. A search for the terms ‘dog or canine’ yielded 1,300,714 articles. A search for the terms ‘hind limb or pelvic limb or extremity’ yielded 438,229 articles. A search for the terms ‘femur or femoral’ and ‘tibia or tibial’ yielded 431,116 and 209,343 articles, respectively. The combination of these search results narrowed the number of articles down to 663. The numbers of these articles were narrowed down to 403 using the filters ‘veterinary medicine, medicine, and agricultural and biological science’. The same procedure was performed for PubMed/Medline and Web of Science databases. The number of new articles from PubMed/Medline was 52, and Web of Science yielded 86 articles; furthermore, 47 articles were added to the list from references and other sources.

### 2.3. Eligibility Criteria

All included articles had to be published in English and articles in other languages were excluded. After excluding the duplicates (n = 35), the titles and abstracts of the selected articles were evaluated, and unrelated articles (n = 485) were excluded. The final evaluation was carried out by reading the full text of the 68 remaining articles.

### 2.4. Study Selection

A large number of hindlimb alignments have been investigated in different studies; however, some of the studies have focused on the accuracy of different methods or tools, as well as intra- and inter-observer agreements. These studies were excluded from this systematic review (n = 39) and only articles that evaluated their results quantitatively were included in the current study. After the exclusion of these studies, 29 articles were included in the systematic review. The articles were assessed separately according to the measured alignments, imaging methods, dog breeds, and the health status of the dogs, and the results were summarized. All of the results were rounded to one decimal place in this review. The exclusion process is explained in Figure 1, according to the PRISMA flow diagram.

## 3. Results

### 3.1. Study Overview

According to the purposes of the study, articles were classified into two main groups. The first group was the articles that focused on reporting standard methods for measurements of femoral and tibial alignments and reporting reference values [16,17,18,19,20,21,22,23,24,25,26]. Eleven articles from a total of 29 articles were included in this group. The second group consisted of studies that compared the femoral and tibial alignments in different dog breeds with and without different orthopedic diseases, such as cranial cruciate ligament (CrCL) rupture, different grades of medial or lateral patellar luxation (MPL or LPL respectively), and osteoarthritis [27,28,29,30,31,32,33,34,35,36,37,38,39,40,41,42,43,44]. Eighteen articles were contained in the second group. The included studies are summarized in Table 1.

### 3.2. Imaging Methods

Different imaging methods have been used to measure the hind limb alignments, such as radiography, computed tomography (CT), and digital photography. In some studies, measurements have been carried out using only one imaging method, though in some studies, different imaging techniques have been compared. The numbers of studies with different imaging methods are shown in Table 2.

### 3.3. Animals

Different dog breeds were evaluated in the included studies. The main aim of these studies was to figure out whether the different breeds had significantly different hind limb alignments. In general, the articles can be divided into studies on small and large breed dogs; however, some studies evaluated a combination of different breeds. Fourteen studies were performed on medium-to-large-breed dogs [16,17,18,19,20,21,22,23,27,28,29,37,39,40]. Eight studies evaluated small-breed dogs [24,30,35,36,38,42,43,44], and 14 studies evaluated a combination of small and large breeds [25,26,31,32,33,34,41]. The number of dogs included in the studies is demonstrated in Figure 2, categorized according to the dog breeds.

### 3.4. Health Status

The dogs in the included studies had different health statuses. Nine studies reported the measured values in healthy dogs [16,17,18,22,23,24,25,26,31]; however, 20 studies investigated dogs with and without different orthopedic diseases, such as CrCL rupture [19,20,21,29,32,33,34,37,38], different grades of MPL or LPL [28,30,35,36,39,40,41,42,43,44], and other orthopedic diseases such as hip dysplasia [27]. The main goal of these studies was to assess the influence of the mentioned orthopedic disease on the hind limb alignments.

### 3.5. Alignments

Femoral and tibial alignments were investigated in the frontal, lateral, and transverse planes. Due to the large number of alignments in the included articles, only the most frequently measured alignments were evaluated in this systematic review. Thirteen femoral alignments and 12 tibial alignments were evaluated in the included articles. The investigated alignments for the femur and tibia are shown in Table 3.

#### 3.5.1. The Femoral Inclination Angle (ICA)

The femoral inclination angle or femoral neck angle is the angle between the long axis of the femoral neck and the anatomical axis of the femur in the frontal plane. ICA transfers biomechanical forces from the femur to the acetabulum. Different methods have been developed for the measurement of the ICA [45,46,47]. The ICA is influenced by the version of the femoral head and neck, in addition to the radiographic positioning. Thus, radiologically exact positioning, as well as the version (ante-, normo-, and retroversion) of the femoral head and neck should be considered as influencing factors. The reported values for ICA in healthy dogs are shown in Table 4.

In 2004, Sarierler [27] reported the values of ICA in dogs with and without hip dysplasia. No significant difference was recorded between dysplastic and non-dysplastic dogs in this study, whereas a significant difference was recorded between Doberman and Labrador, and between Anatolian karabash and the other six breeds (German shepherd, Labrador retriever, golden retriever, Pointer, Doberman pinscher, and Irish setter). Tomlinson et al. [18] measured the ICA values in four large-breed dogs and reported that Rottweilers had significantly higher ICA values than German shepherds, golden retrievers, and Labrador retrievers; furthermore, the ICA values of the golden retrievers were significantly higher than those of German shepherds in that study. In 2009, Mortari et al. [28] reported significantly lower postoperative ICA values for dogs with grades 2 and 3 MPL (lateral retinacular overlap and wedge recession sulcoplasty for dogs with grade 2 MPL and lateral retinacular overlap, wedge recession sulcoplasty, release of the quadriceps muscle, and tibial tuberosity transposition for dogs with grade 3 MPL) in comparison with preoperative values. Soparat et al. [30], found no significant difference in ICA values in Pomeranians with and without MPL. The same results were reported by Olimpo et al. [35], for different small-breed dogs; Yasukawa et al. [36], for Toy Poodles; Lusetti et al. [39], for English bulldogs; Newman and Voss, [40], for English Staffordshire bull terriers; Perry et al. [41], for a combination of different dog breeds; Phetkaew et al. [43], for Chihuahuas; and Žilinčík et al. [44], for Yorkshire terriers. Perry et al. [41], reported a significant association between increased ICA values and postoperative complications in dogs that had undergone MPL surgery. For every increased degree of ICA, the odds of complications increased by 0.1, as recorded in that study. Kara et al. [26], investigated ICA values in small-to-medium-breed dogs using 3D images reconstructed from CT scans and reported no correlation between ICA and aLDFA or aCdDFA in normal canine femora. In another study, Garnoeva et al. [42], reported that higher ICA values were recorded for dogs with grades 2 and 3 MPL in comparison with healthy dogs, which was contrary to the previous studies. The ICA values in dogs with different grades of MPL are shown in Table 5.

#### 3.5.2. The Anatomical Lateral Proximal Femoral Angle (aLPFA) and the Mechanical Lateral Proximal Femoral Angle (mLPFA)

The aLPFA is the angle in the proximal lateral side of the femur in the frontal plane between the anatomical axis of the femur and the proximal joint reference line [1,18]. The mLPFA is the angle in the proximal lateral side of the femur in the frontal plane between the mechanical axis of the femur and the proximal joint reference line [1,18]. The importance of aLPFA and mLPFA is its use in evaluating the anatomical structure of the proximal femur, especially in the case of fracture, and in assessing the healing process [18]. The reported aLPFA and mLPFA values in healthy dogs are shown in Table 6.

Various studies have reported aLPFA and mLPFA values in sound dogs. Tomlinson et al. [18], reported a significant difference between the shape of the greater trochanter and femoral head among Labrador retrievers, golden retrievers, German shepherds, and Rottweilers. Labrador retrievers had significantly higher aLPFA and mLPFA values than the other three dog breeds in this study; however, German shepherds had significantly higher aLPFA and mLPFA values than golden retrievers and Rottweilers. No significant difference was reported in mLPFA for the golden retrievers and Rottweilers in that study. The radiographic positioning of the dogs is an important factor in achieving accurate measurements. Radiographs are susceptible to positioning error, and the risk of positioning error rises in dogs with bone deformities, such as severe grades of patellar luxation [28]. The radiographic positioning of the femur can influence the relative position of the greater trochanter, which is an important landmark used to draw the proximal joint orientation line on the femur [18]. In 2016, Olimpo et al. [35], in a radiographic study, reported no significant difference in aLPFA and mLPFA between dogs with and without different grades of MPL. Yasukawa et al. [36], reported the same results with radiography and CT scans for healthy and MPL (grade 2 and 3)-affected Toy Poodles. These results were confirmed by Lusetti et al. [39], for purebred English bulldogs with and without MPL using the CT method, and by Garnoeva et al. [42], for sound and MPL-affected small-breed dogs. In 2018, Phetkaew et al. [43] reported a significant difference in aLPFA and mLPFA values between CT scans and craniocaudal and caudocranial radiographs in healthy Chihuahuas. Žilinčík et al. [44], reported that the mean aLPFA values in Yorkshire terriers with grade 4 MPL were significantly lower than those in other MPL groups. The aLPFA and mLPFA values observed in dogs with MPL are shown in Table 7.

#### 3.5.3. The Anatomical Lateral Distal Femoral Angle (aLDFA), Mechanical Lateral Distal Femoral Angle (mLDFA), and Femoral Varus Angle (FVA)

The aLDFA and mLDFA are the lateral distal angles of the femur, between the distal joint orientation line and the anatomical and mechanical axis of the femur in the frontal plane, respectively [1,18]. The FVA is an angle between the distal anatomical axis of the femur in the frontal plane and the perpendicular line drawn to the intersection of the anatomical axis and distal joint orientation line [1,16]. aLDFA, mLDFA, and FVA are important alignments used to evaluate distal femoral deformities. The incidence of varus or valgus deformities in the distal part of the femur is greater than that in the proximal femur [14,18], and these deformities may be one of the predisposing factors for patellar luxation in dogs, although the exact amount of femoral distal varus or valgus deformities that cause MPL or LPL is unknown [18]. The aLDFA, mLDFA, and FVA values reported in healthy dogs are shown in Table 8.

In 2007, Tomlinson et al. [18] reported no significantly different aLDFA and mLDFA values between Labrador retriever, golden retriever, and Rottweiler dogs, but German shepherds had significantly lower values than the other three large breeds. Later in 2012, Soparat et al. [30] reported significantly greater aLDFA, mLDFA, and FVA values for Pomeranians with grade 3 MPL in comparison with lower grades of MPL and healthy Pomeranians. These findings were confirmed by Yasukawa et al. [36], who reported significantly greater aLDFA, mLDFA, and FVA values for Toy Poodles with grade 4 MPL. The study performed by Olimpo et al. [35], confirmed these findings too and showed a significantly higher aLDFA for small-breed dogs with grade 4 MPL in comparison with lower grades (1–3 MPL) and healthy dogs. The same results were reported by Lusetti et al. [39], for English bulldogs regarding aLDFA and mLDFA, and by Garnoeva et al. [42], for aLDFA, mLDFA, and FVA in small-breed dogs with MPL. Newman and Voss, [40], reported increased aLDFA values in English Staffordshire bull terriers with MPL but this increase was not statistically significant in that study. In 2018, Phetkaew et al. [43] reported that mLDFA and aLDFA values were related to the severity of MPL in Chihuahuas, which was confirmed by Žilinčík et al. [44], who reported significantly greater aLDFA and FVA values for Yorkshire terriers with grade 4 MPL in comparison with other grades of MPL and healthy Yorkshire terriers. The aLDFA, mLDFA, and FVA values reported in dogs with different grades of MPL are shown in Table 9.

#### 3.5.4. Quadriceps Angle (Q angle)

The Q angle is the angle between the long axis of the rectus femoris muscle and the patellar ligament. It represents the force generated by the quadriceps muscle [22,25,48]. Łojszczyk-Szczepaniak et al. [22] reported standard values in healthy German shepherd dogs, which was larger than those reported by Kaiser et al. [48], for the sound dogs. The Q angle has been reported to be increased in dogs with MPL [48]. In 2009, Mortari et al. [28] evaluated pre- and postoperative values of the Q angle in dogs with different grades of MPL that underwent reconstructive surgery. According to the severity of the MPL, one or a combination of the different surgical methods had been performed on these dogs, including lateral retinacular overlap, wedge recession sulcoplasty, medial desmotomy, release of the quadriceps muscle, and tibial tuberosity transposition. The authors reported a significant pre-operative difference between the dogs with grades 1 and 3 MPL and between the dogs with grades 2 and 3 MPL. The postoperative Q angle was decreased (24.13%) in dogs with grade 3 MPL; nevertheless, the difference between pre- and postoperative Q angles was not statistically significant. In 2017, Pinna and Romagnoli [25] reported a significantly higher Q angle for sound small-breed dogs (below 15 kg) in comparison with large breeds. A study performed by Garnoeva et al. [42], confirmed that the Q angle in dogs with different grades of MPL was significantly higher than the Q angle in non-affected dogs. These results showed that the Q angle increased in dogs with MPL and confirmed previous studies. The values of the Q angles are shown in Table 10.

#### 3.5.5. The Anatomical Caudal Proximal Femoral Angle (aCdPFA), Mechanical Caudal Proximal Femoral Angle (mCdPFA), Anatomical Caudal Distal Femoral Angle (aCdDFA), Mechanical Caudal Distal Femoral Angle (mCdDFA), and Procurvatum Angle (PA)

aCdPFA and mCdPFA are the angles between the anatomical and mechanical femoral axis in the sagittal plane, respectively, and the femoral neck axis [1,36,43]. aCdDFA and mCdDFA are the angles between the anatomical and mechanical femoral axis in the sagittal plane, respectively, and the axis of the distal femur [1,36,43]. PA is defined as the angle between the proximal and distal anatomical axis of the femur in the sagittal plane [36,43]. Proximal and distal femoral angles in the sagittal plane have been evaluated in few articles. Only three studies have assessed the femoral alignments in the sagittal plane. Yasukawa et al. 2016 [36] evaluated all of these alignments in Toy Poodles with and without MPL (grades 2 and 4) using radiography and CT scanning. The authors reported no significant difference between healthy and affected dogs. Phetkaew et al. 2018 [43], evaluated aCdPFA, aCdDFA, and PA in Chihuahuas, and reported a significant difference in CT imaging and both craniocaudal and caudocranial radiographs in healthy stifles for the measured values. Based on the CT scans, the aCdPFA value was related to the severity of MPL in Chihuahuas. The results showed that the aCdPFA value significantly decreased in grades 2, 3, and 4 MPL. The results reported by Kara et al. 2018 [26], on normal femoral specimens showed an inverse correlation between AA and aCdDFA; however, no significant difference between male and female dogs was reported for aCdDFA in that study. The reported values for the femoral alignments in the sagittal plane are shown in Table 11.

#### 3.5.6. Angle of Anteversion (AA)

The AA is defined as the relative position of the femoral neck to the femoral condyles in the transverse plane and is used to evaluate the torsion of the femur [16]. Femoral torsion is defined as the rotation of the femur around its anatomical axis. Femoral torsion and femoral ante- or retroversion have two separate definitions but most of the time are considered equivalent in the literature because of the difficulty of the separate identification of these values [16]. In 1973, Nunamaker et al. [49] reported a method for the measurement of AA on axial view radiographs. Radiography is the easiest method used to measure femoral torsion; however, radiographs are vulnerable to positioning errors that may affect the measured values. Given that CT and magnetic resonance imaging (MRI) are reported as gold standards in human medicine [16], Dudley et al. [16], described a CT technique for the determination of AA in sound dogs and compared this technique with previously used standard radiography and anatomical preparation. The results showed no significant difference between the three methods. The reported AA values in healthy dogs are shown in Table 12.

In a study performed by Olimpo et al. [35], no significant difference was reported for AA in small-breed dogs with and without MPL using radiographs. The same results were reported by Lusetti et al. [39], who reported no significant difference in AA values in English bulldogs with and without MPL using CT scans, and by Phetkaew et al. [43], who reported no significant difference in AA values in Chihuahuas with and without MPL using CT scans. Contrary to these studies, Yasukawa et al. [36], reported a significantly lower AA values with CT imaging for Toy Poodles with grade 4 MPL compared to the grade 2 MPL and healthy Toy Poodles. In 2017, Newman and Voss [40] evaluated the AA (overall), proximal AA (PAA), and distal AA (DAA) in English Staffordshire bull terriers with and without congenital MPL using CT scans and reported significantly decreased AA and DAA values in dogs with grades 2 and 3 MPL, which aligned with the results of Yasukawa et al. However, a radiographic study performed by Žilinčík et al. [44], on Yorkshire terriers with and without MPL showed that dogs with grade 4 MPL had significantly lower AA values in comparison with other groups. In a study performed by Kara et al. [26], a weak inverse correlation was reported between AA and aCdDFA; however, a weak positive correlation was reported between AA and aLDFA in that study. The AA values reported in dogs with orthopedic diseases are shown in Table 13.

#### 3.5.7. The Mechanical Medial Proximal Tibial Angle (mMPTA) and Mechanical Medial Distal Tibial Angle (mMDTA)

The mMPTA is an angle in the medial side of the tibia in the frontal plane, between the mechanical axis and the proximal joint orientation line, which is represented by a line passing through the distal points of the concavities of the medial and lateral tibial condyles [1,19,20]. The mMDTA is an angle in the medial side of the tibia in the frontal plane, between the mechanical axis and distal joint orientation line, which is represented by a line passing through the proximal points of the medial and lateral concavities of the tibial cochlea [1,19,20]. Standard values of mMPTA and mMDTA were reported by Olimpo et al. [35], and Garnoeva et al. [42], in small-breed dogs; by Lusetti et al. [39], in English bulldogs; and by Phetkaew et al. [43], in Chihuahuas. The reported values are shown in Table 14.

Fuller et al. [32], compared the mMPTA and mMDTA values in dogs with bilateral and unilateral CrCL rupture, and reported no statistical difference between these groups. In 2016, Olimpo et al. [35] investigated mMPTA and mMDTA in small breed dogs with and without MPL and reported significantly greater mMPTA values for the dogs with grade 4 MPL. These findings were confirmed by Garnoeva et al. [42]; furthermore, higher mMDTA values were reported for dogs with grade one MPL, compared to healthy dogs in that study. Contrary to these findings, some studies reported no significant difference in mMPTA and mMDTA in Toy Poodles [36] and English bulldogs [39] with and without MPL. In a study performed on Chihuahuas with and without MPL, the mMPTA and mMDTA values measured using radiography differed significantly from those measured using CT imaging [43]. The reported values for mMPTA and mMDTA in dogs with MPL and CrCL disease are shown in Table 15.

#### 3.5.8. The Tibial Plateau Angle (TPA), Diaphyseal Proximal Tibial Angle (DPA), Mechanical Caudal Proximal Tibial Angle (mCdPTA), Mechanical Cranial Distal Tibial Angle (mCrDTA), Z Angle, and Relative Tibial Tuberosity Width (rTTW)

The TPA is the angle between the proximal joint orientation line in the sagittal plane and the line drawn perpendicular to the mechanical bone axis at the level of the joint orientation line [32,33,34,37]. The DPA is the angle in the sagittal plane between the diaphyseal tibial axis and a straight line passing through the cranial aspect of the medial tibial condyle and the midpoint of the perpendicular line drawn from the distal aspect of the tibial crest to the diaphyseal tibial axis [17]. The mCdPTA is the angle between the mechanical axis of the tibia in the sagittal plane and the proximal joint orientation line in the caudal aspect of the tibia [1,21]. The mCrDTA is the angle between the mechanical axis of the tibia in the sagittal plane and the distal joint orientation line in the cranial aspect of the tibia [1,21]. The Z angle is the angle between the mechanical axis of the tibia in the sagittal plane and a straight line connecting the most cranial point of the tibial tuberosity with the midpoint between two tibial intercondylar tubercles [31,33]. Measurement of the rTTW is based on the identification of the most cranial (A) and distal (B) points of the tibial plateau in the sagittal plane, the most proximal point of the tibial crest (C), and the cross point of the circle with center B and radius AB. Relative tibial tuberosity width is the ratio CD/AB [24,31,33].

In 2006, Osmond et al. [17] investigated the morphology of the proximal portion of the tibia in dogs with and without CrCL rupture using radiographs. The results showed that the dogs with CrCL rupture had significantly higher DPA values than healthy dogs. In 2008, Dismukes et al. [21] described a method for determining mCdPTA and mCrDTA in the sagittal plane and reported no difference for measured angles between Labrador retrievers and non-Labrador retrievers with CrCL disease. Ragetly et al. [29], reported that a combination of measured TPA and the femoral anteversion angle on radiographs was optimal for distinguishing predisposed and non-predisposed limbs for CCL disease in Labrador retrievers. The authors considered healthy hind limbs to be non-predisposed to CrCL disease and the contralateral limbs with CrCL disease to be predisposed limbs. The measured values were increased in predisposed limbs. In 2013, Vedrine et al. [31] investigated tibial conformation in healthy Labrador retrievers and Yorkshire terriers and compared measured alignments between these two breeds. TPA, Z angle, DPA, and rTTW were measured and reference values were reported. The authors reported a significant effect of breed on the measured values. Labrador retrievers had a lower TPA, Z angle, DPA, and rTTW than Yorkshire terriers. The DPA was correlated with TPA, Z angle, and rTTW; in addition, the TPA was also correlated with the Z angle in that study. Sabanci and Ocal, [23], compared the lateral and medial TPA in sound dogs using radiography and photography. The normal radiographic TPA was compared with the lateral and medial photographic TPA in this study. A significant difference was recorded between medial and lateral TPA using the photographic method; furthermore, the difference in the photographic medial TPA between male and female dogs was significant. Fuller et al. 2014 [32], evaluated the TPA, mCdPTA, and mCrDTA in dogs with bilateral and unilateral CrCL rupture and reported no static difference between the groups; therefore, the mentioned angles were not considered to be a risk factor for subsequent contralateral CrCL rupture. In 2015, Witte [24] assessed proximal tibial alignments in small-breed dogs and reported that the TPA, DPA, Z angle, and rTTW in the small-breed dogs were higher than those reported for large-breed dogs previously. Aertsens et al. [33], investigated the TPA, Z angle, and rTTW in small- and large-breed dogs with CrCL disease. The results showed that the small-breed dogs with CrCL disease had a greater TPA and Z angle than large-breed dogs with CrCL disease. A strong correlation was found between the TPA and the Z angle. Sex and neutered status influenced the TPA and Z angle values, whereas no significant effect was observed on the rTTW values. Su et al. [34], compared TPA in small- and large-breed dogs. The measurements were performed on radiographs of dogs with and without CrCL disease. The results showed that small-breed dogs had higher TPA values than large-breed dogs. In 2016, Olimpo et al. [35] reported that the TPA values in small-breed dogs with grade 4 MPL were significantly greater than those in other groups; however, the mCdPTA values in healthy dogs were significantly lower than those in dogs with different grades of MPL. No significant difference was reported for mCrDTA between the groups. Yasukawa et al. [36], investigated mCrPTA, mCrDTA, TPA, Z angle, and rTTW in Toy Poodles with and without MPL using radiography and CT and reported no significant difference among the healthy and affected dogs. In 2017, Guénégo et al. [37] evaluated the TPA, rTTW, and Z angle in dogs at low risk of CrCL rupture and in predisposed dogs. A significant difference was recorded between the control group and the CrCl group for all of the measured alignments. In the CrCL group, rTTW was significantly lower than those in the control group but TPA and Z angle were significantly increased in the CrCL group compared to the control group. Janovec et al. [38], reported that the dogs with CrCL rupture had significantly greater sTPA values (TPA as described by Slocum and Slocum [50]) and relative body weight than healthy dogs. Lusetti et al. [39], reported no significant difference in mCdPTA and mCdDTA in English Bulldogs with and without MPL with CT, whereas Garnoeva et al. [42], reported greater mCrPTA values in healthy dogs compared to dogs with grade 3 MPL. Furthermore, a significant difference was recorded in the mCdDTA of healthy dogs and dogs with grade 2 MPL in that study. Phetkaew et al. [43], reported a significant difference between radiographs and CTs in terms of the mCrDTA in Chihuahuas with grade 2 MPL. The evaluated tibial alignments in the sagittal plane are summarized in Table 16.

## 4. Discussion

This systematic review was carried out on the previously performed studies to assess the femoral and tibial conformations in dogs. The main goals of this study were to evaluate each alignment separately to report the reference values for healthy dogs and to report the differences between healthy and diseased dogs. These values could give veterinarians an important overview in order to comprehend the anatomical variations of different dog breeds.

Investigations of ICA in most of the articles showed the same and homogenous results. Studies on healthy dogs showed a significant difference between some of the breeds but not all of them. Most of these differences were identified between large-breed dogs [18,27], whereas no study compared ICA values between healthy small-breed dogs. According to the studies reviewed here, no significant difference was recorded between the dogs with and without MPL, or between dysplastic and non-dysplastic dogs [27,30,35,36,39,40,43,44]. Only one study reported higher ICA values for small-breed dogs with grades 2 and 3 MPL [42]. It can be deduced that the ICA does not differ significantly in dogs with and without MPL or dysplastic and non-dysplastic dogs. No studies investigated the relationship between ICA and other orthopedic diseases.

The measured aLPFA and mLPFA values differed between some of the healthy large-breed dogs [18], whereas in most of the articles no significant difference was reported between healthy small breeds [35,36,39,42]. In one study, a significant difference was recorded between radiographic and CT methods in healthy Chihuahuas [43]. Further investigations are required to assess the influence of breeds on the measured values, especially in large breeds. In most of the articles, no significant differences were reported in aLPFA and mLPFA values between dogs with and without MPL [35,36,39,42]. Only a significantly lower aLPFA value was reported in Yorkshire terriers with grade 4 MPL [44]. Furthermore, no significant difference was recorded between dogs with CrCL rupture [20]. According to these results, it could be supposed that the reference values of these alignments may differ between some breeds, but MPL and CrCL disease do not influence the aLPFA and mLPFA values.

A significant difference was recorded for some of the healthy large breed dogs in regard to aLDFA and mLDFA values [18], whereas no studies investigated distal femoral alignments in healthy small breeds alone, and all studies were undertaken between healthy and diseased small-breed dogs. Contrary to the proximal femoral alignments, aLDFA, mLDFA, and FVA were correlated to the severity of the MPL and higher values were recorded for small-breed dogs with higher grades of MPL, such as Pomeranian, Toy Poodle, English bulldog, English Staffordshire bull terrier, Chihuahua, and Yorkshire terrier [30,35,36,39,40,42,43,44]. No studies evaluated these alignments in large-breed dogs with and without orthopedic diseases; however, studies were focused on small-breed dogs with MPL, and the influence of other orthopedic diseases is unknown. Few studies investigated Q angles in dogs. Only two studies reported values in healthy dogs [22,25]. According to these articles, significantly higher Q angles were reported for healthy small breeds [25] but more studies are needed to confirm these findings. The measured Q angles compared between healthy dogs and dogs with and without MPL showed a correlation between the severity of MPL and the Q angle [28,42]. In dogs with higher grades of MPL, higher Q angles were recorded [28,42]. No studies investigated the influence of other orthopedic diseases on the Q angle.

Only a small numbers of articles were focused on proximal and distal femoral alignments in the sagittal plane. No significant difference was recorded between sound male and female dogs [26]. Measurements of aCdPFA, mCdPFA, aCdDFA, mCdDFA, and PA in healthy and MPL-affected dogs resulted in two different findings. In one study, no significant difference was recorded between healthy and MPL-affected Toy Poodles [36], whereas in another study significantly decreased aCdPFA values were reported in Chihuahuas with grades 2, 3, and 4 MPL [43]. According to these findings, an accurate deduction cannot be expected, and further investigation should be performed to reach an accurate result.

No significant difference was reported between measurements of the AA with CT, radiography, and digital photography (cadaveric specimens) in healthy dogs [16]. In a 3D morphometric study, a weak correlation was seen between AA and aLDFA in healthy dogs; however, a weak inverse correlation was recorded between AA and aCdDFA in healthy dogs [26]. Evaluation of the AA in healthy and MPL-affected dogs yielded variable results—some articles reported no significant difference between small breeds such as English bulldogs and Chihuahuas with and without MPL [35,39,43], whereas other studies showed significantly lower AA values in Toy Poodles, English Staffordshire bull terriers, and Yorkshire terriers with higher grades of MPL [36,40,44]. The effect of other orthopedic diseases on AA was not investigated; furthermore, no studies evaluated AA values in healthy and diseased large-breed dogs.

According to a study performed on healthy and MPL affected Chihuahuas, a significant difference was detected between radiographic and CT measurements of mMPTA and mMDTA in healthy Chihuahuas [43]. Two studies evaluated mMPTA and mMDTA in dogs with CrCL rupture and reported no significant difference between these dogs [19,32]. No significant difference was found between healthy and MPL-affected Toy Poodles and English bulldogs in regard to mMPTA and mMDTA [36,39], whereas in other studies performed on a combination of small breeds, higher mMPTA values were recorded for the dogs with MPL [35,42]. No studies investigated mMPTA and mMDTA in large-breed dogs with orthopedic diseases. These findings are not consistent, and therefore in order to obtain a firm conclusion about mMPTA and mMDTA, more investigations are needed.

Assessments of the tibial alignments in the sagittal plane yielded various outcomes. Studies performed on healthy dogs reported higher TPA, DPA, Z angle, and rTTW values for small breeds in comparison with large breeds [24,31]. Studies performed on dogs with and without CrCL disease showed a significantly higher TPA for dogs with CrCL disease [17,34,37,38]; however, the range of the TPA values in small-breed dogs with CrCL disease was higher than those of large breeds with CrCL disease [33,34]. Contrary to these findings on small breeds with CrCL disease, only one study evaluated large breeds with CrCL disease and reported no significant difference for large breeds with different grades of CrCL disease [32]. Two different results were reported in the articles for dogs with MPL regarding the TPA. In one study, higher TPA values were recorded for a combination of small breed dogs with grade 4 MPL [35], whereas in another study no difference was reported comparing Toy Poodles with and without MPL [36]. TPA was correlated to the Z angle as well [33].

The studies performed on DPA reported significantly higher values in healthy small-breed dogs [31]. Additionally, DPA was correlated with TPA, Z angel, and rTTW in that study. A significant difference was recorded between the dogs with and without CrCL diseases. Higher DPA values were recorded for dogs with CrCL diseases [17].

The Z angle significantly differed between healthy small- and large-breed dogs, and higher values were recorded for small breeds such as Yorkshire terriers [31]; however, dogs with CrCL rupture had higher Z angles in comparison with sound dogs [33,37]. With the exception of dogs with CrCL rupture, the Z angle did not differ between Toy Poodles with and without MPL [36]. The results reported for rTTW were not homogenous in the included studies—in one study, significantly lower values were recorded for CrCL-ruptured dogs [37], whereas no significant difference was reported in another study comparing small-breed dogs with and without CrCL rupture [38]. Furthermore, no significant difference was recorded between MPL-affected and healthy dogs [36].

The results of the several studies which investigated mCrPTA, mCrDTA, mCdPTA, and mCdDTA showed no significant difference between healthy dogs and dogs affected with MPL or CrCL rupture [21,32,36,39]. However, in one study, a lower mCrPTA was reported for small breeds with grade 3 MPL, and a significantly higher mCdDTA was reported for dogs with grade 2 MPL in comparison with healthy dogs [42]. In another study, a significant difference was reported between radiography and computed tomography for Chihuahuas with grade 2 MPL regarding the mCrDTA [43].

This systematic review had several limitations. We tried to compile and categorize the articles and results according to our study aims. Considering that each study was performed in different circumstances with different tools, it was not possible to reach a homogenous conclusion in some cases. Some studies evaluated only one dog breed, whereas other studies evaluated a combination of different breeds and the results were reported as an average of those breeds. The measured values were normally reported as means ± standard deviation, but in some cases the results were reported as medians (minimum–maximum). The second group of studies consisted of dogs with and without different orthopedic diseases. In most studies, dogs with and without MPL were investigated, and only a few studies investigated other diseases, such as CrCL disease. Most of the assessments comparing between healthy dogs and dogs with CrCL disease were limited to the tibial alignments, especially in the sagittal plane. The number of studies that investigated large-breed dogs with and without orthopedic diseases was lower than the number of studies on small breeds; therefore, monotone comparison between small and large breed dogs was not possible. The comparison of the values reported by two or more studies with the same methods or tools was another study limitation in this review, as we did not perform a meta-analysis in this systematic review. Therefore, we were not able to determine if these differences were statistically significant or not. Other study limitations were the low sample size and various sample types in the literature. In some of the studies, investigations were performed on cadaveric specimens, whereas in other studies dogs were investigated. As reported previously, the correct positioning of the animals is very important in order to achieve proper radiographic measurements; therefore, the results of cadaveric studies may differ from clinical studies. Further investigations are needed to clarify this issue. Another limitation was the high amount of information in the articles, which makes it difficult to summarize all the reported results and information; thus, the measurement methods, intra- and inter-observer agreements, and other details were not included in this review, and only frequently used alignments were reported.

## 5. Conclusions

Distal femoral alignments in the frontal plane (aLDFA, mLDFA, FVA, and Q angle) and tibial alignments in the frontal plane (mMPTA, mMDTA, and TPA) corresponded to the severity of the MPL. The difference between affected and non-affected dogs with CrCL rupture was limited to the proximal tibial alignments in the sagittal plane, including TPA, Z angle, and DPA, which shows the significance of proximal tibial conformations in dogs with CrCL rupture. Statistically, significant differences were recorded between some of the dog breeds for different angles, although these results were not valid between other breeds. Most of the differences were recorded between large breeds or between large and small breeds, whereas no comparisons were made between each small-breed dog in isolation. The number of articles that evaluated the influence of body size, anatomy, or breed on the measured alignments was low, thus no strong conclusion could be drawn in this regard. Further investigations should be performed to determine the influence of breed on hindlimb conformations and the occurrence of the related orthopedic diseases.

## Figures and Tables

**Figure 1 animals-11-01804-f001:**
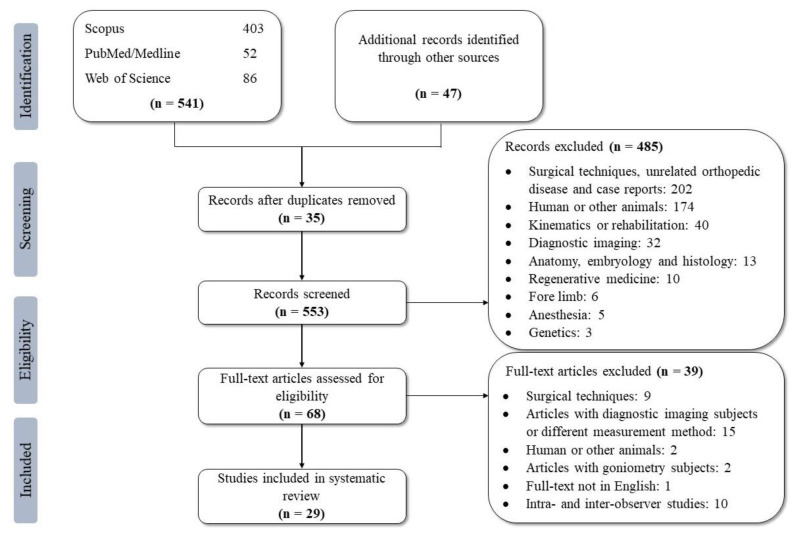
PRISMA [15] flow diagram depicting the number of identified articles and the exclusion process.

**Figure 2 animals-11-01804-f002:**
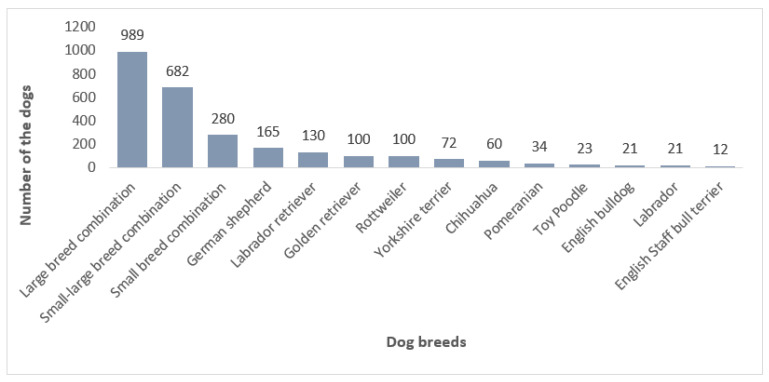
Classification of the dog breeds in the included articles.

**Table 1 animals-11-01804-t001:** Overview of the studies included in the systematic review.

Study	Study Group ^1^	Dog Breed	Number of Dogs	Health Status	Imaging Method	Alignments ^2^
Aertsens et al. 2015 [33]	2	Small and large	104	CrCL rupture ^3^	Radiography	TPA, rTTW, Z angle
Dismukes et al. 2007 [19]	1	Large	105	CrCL rupture	Radiography	mMPTA, mMDTA
Dismukes et al. 2008a [20]	1	Medium to large (Cadaver)	52	CrCL rupture	Radiography	mLPFA, mLDFA, mMPTA, mMDTA, mTFA, mMTTA, MAFA, MAMTA, SMAD, TMAD
Dismukes et al. 2008b [21]	1	Large (Cadaver)	150	CrCL rupture	Radiography	mCrDTA, mCdPTA
Dudley et al. 2006 [16]	1	Medium to large (Cadaver)	9	Sound	CT ^4^ and Radiography	FVA, AA
Fuller et al. 2014 [32]	2	Different breeds	106	CrCL rupture	Radiography	TPA, mCaPTA, mCrDTA, SPA, FPA, mMPTA, mMDTA
Garnoeva et al. 2018 [42]	2	Small	87	Sound and MPL ^5^	Radiography	aLPFA, mLPFA, aLDFA, mLDFA, FVA, IFA, Q angle
Guénégo et al. 2017 [37]	2	Large	274	Sound and CrCL rupture	Radiography	AMA, TPA, rTTW, Z angle
Janovec et al. 2017 [38]	2	Small	133	Sound and CrCL rupture	Radiography	sTPA, nTPA, PTTA, TPL/DTW, PTW, DTW, PTW/TPL, nTPA/PTW, rTTW, nTPA/PTWq, rBW
Kara et al. 2018 [26]	1	Different breeds (Cadaver)	75 (femora)	Sound	CT	aLDFA, CDFA, AVA, NSA
Łojszczyk-Szczepaniak et al. 2014 [22]	1	German shepherd	65	Sound	Radiography	Q angle
Lusetti et al. 2017 [39]	2	English bulldog	21	Sound and MPL	CT	aLPFA, aLDFA, mLPFA, mLDFA, ICA, AA, mMPTA, mMDTA, mCdDTA, mCdPTA, TTA
Mortari et al. 2009 [28]	2	Small to medium	18	MPL	Radiography	ICA, Norberg angle, Q angle, FVA
Newman and Voss, 2017 [40]	2	English Staffordshire bull terrier	12	Sound and MPL	CT	ICA, AA, DAA, PAA, FVA, TVA, TTA, aLDFA, FCT, TV, TT
Olimpo et al. 2016 [35]	2	Small	48	Sound and MPL	Radiography	aLPFA, aLDFA, mLPFA, mLDFA, Femoral Anteversion, mCaPTA, mCrDTA, mMPTA, mMDTA, TPA
Osmond et al. 2006 [17]	1	Large	67	Sound	CT and Radiography	TPS, TPO, DPA
Perry et al. 2017 [41]	2	Different breeds	73	MPL	Radiography	FVA, aLDFA, mLDFA, ICA
Phetkaew et al. 2018 [43]	2	Chihuahua	60	Sound and MPL	CT and Radiography	aLPFA, aLDFA, mLPFA, mLDFA, ICA, mMPTA, mMDTA, PA, mCdPTA, mCrDTA, aCdPFA, aCdDFA
Pinna and Romagnoli, 2017 [25]	1	Small to large	160	Sound	Radiography	Q Angle
Ragetly et al. 2011 [29]	2	Labrador	21	Sound and CrCL rupture	CT and Radiography	TPA, FAA (FNA angle)
Sabanci and Ocal, 2014 [23]	1	Medium to large	90	Sound	Radiography and Photography	TPA
Sarierler, 2004 [27]	2	Medium to large	242	Dysplastic and non-dysplastic	Radiography	ICA
Soparat et al. 2012 [30]	2	Pomeranian	34	Sound and MPL	Radiography	ICA, FVA, aLDFA, mLDFA
Su et al. 2015 [34]	2	Different breeds	146	Sound and CrCL rupture	Radiography	TPA
Tomlinson et al. 2007 [18]	1	Large	400	Sound	Radiography	aLPFA, mLPFA, aLDFA, mLDFA, ICA
Vedrine et al. 2013 [31]	2	Labrador retriever and Yorkshire terrier	60	Sound	Radiography	TPA, PTA, Z angle, DPA, rTTW
Witte, 2015 [24]	1	Small	12	Sound	Radiography	TPA, DPA, PTA, Z angle, rTTW
Yasukawa et al. 2016 [36]	2	Toy Poodle	23	Sound and MPL	CT and Radiography	aLPFA, mLPFA, aLDFA, mLDFA, FVA, IFA, PA, mCdPFA, mCdDFA, aCdPFA, aCdDFA, AA, FFA, mMPTA, mMDTA, mCrPTA, mCrDTA, TPA, Z angle, rTTW, TTA, MDTT/PTW
Žilinčík et al. 2018 [44]	2	Yorkshire terrier	42	Sound and MPL	Radiography	aLPFA, aLDFA, AA, FIA (ICA), FVA

**^1^***Study group*: 1. Studies that reported standard values. 2. Studies that compared the values in dogs with and without different orthopedic diseases. **^2^***Alignments:* AA (angle of anteversion or femoral torsion angle); AVA (angle of anteversion); aCdDFA (anatomical caudal distal femoral angle); aCdPFA (anatomical caudal proximal femoral angle); aLDFA (anatomical lateral distal femoral angle); aLPFA (anatomical lateral proximal femoral angle); AMA-angle (anatomical-mechanical axis angle); CDFA (caudal distal femoral angle); DAA (distal angle of anteversion); DPA (distal tibial axis/proximal tibial axis angle or diaphyseal proximal tibial angle); DTW (diaphyseal tibial width); FAA (femoral anteversion angle); FCT (femoral trochanteric angle); FIA (femoral inclination angle); FNA angle (femoral neck anteversion angle); FPA (frontal plane alignment); FVA (femoral varus angle); ICA (angle of inclination); IFA (intercondylar fossa angle); MAFA (mechanical axis—femur angle); MAMTA (mechanical axis—metatarsal angle); mCaPTA (mechanical cranial proximal tibial angle); mCdDFA (mechanical caudal distal femoral angle); mCdDTA (mechanical caudal distal tibial angle); mCdPFA (mechanical caudal proximal femoral angle); mCdPTA (mechanical caudal proximal tibial angle); mCrDTA (mechanical cranial distal tibial angle); mCrPTA (mechanical cranial proximal tibial angle); mLDFA (mechanical lateral distal femoral angle); mLDTA (mechanical lateral distal tibial angle); mLPFA (mechanical lateral proximal femoral angle); mLPTA (mechanical lateral proximal tibial angle); mMDTA (mechanical medial distal tibial angle); MDTT/PTW (ratio of the medial distance of tibial tuberosity to the proximal tibial width); mMPTA (mechanical medial proximal tibial angle); mMTTA (mechanical metatarsotibial angle); mTFA (mechanical tibiofemoral angle); nTPA (tibial plateau angle, Inauen and colleagues’ method); NSA (neck-shaft angle); PA (procurvation angle); PAA (proximal angle of anteversion); PTTA (proximal tibial tuberosity angle); PTW (proximal tibial width); PTWq (proximal tibial width quotient); Q angle (quadriceps angle); rBW (relative body weight); rTTW (relative tibial tuberosity width); sTPA (tibial plateau angle, Slocum and Slocum method); SMAD (stifle mechanical axis deviation); SPA (sagittal plane alignment); TMAD (tarsal mechanical axis deviation); TPA (tibial plateau angle); TPL (tibial plateau length); TPO (tibial plateau orientation); TPS (tibial plateau slope); TTA (tibial torsion angle); TT (tibial torsion); TV (tibial valgus); TVA (tibial valgus angle); Z angle. **^3^**
*CrCL rupture*: cranial cruciate ligament rupture. **^4^**
*CT*: computed tomography. **^5^**
*MPL*: medial patellar luxation.

**Table 2 animals-11-01804-t002:** Classification of the included articles, according to the imaging method.

Imaging Method	Number of Articles	Reference Numbers
Radiography	20	[18,19,20,21,22,24,25,27,28,30,31,32,33,34,35,37,38,41,42,44]
Computed tomography	3	[26,39,40]
Radiography and computed tomography	5	[16,17,29,36,43]
Radiography and photography	1	[23]

**Table 3 animals-11-01804-t003:** Included femoral and tibial alignments in the frontal, sagittal, and transverse planes.

Bone	Plane	Alignments
Femur	Frontal	ICA or FNA, aLPFA, mLPFA, aLDFA, mLDFA, FVA, Q angle
	Sagittal	aCdPFA, mCdPFA, aCdDFA, mCdDFA, PA
	Transverse	AA
Tibia	Frontal	mMPTA, mMDTA
	Sagittal	TPA, DPA, Z angle, rTTW, mCdPTA, mCrDTA

AA (angle of anteversion or femoral torsion angle), aCdDFA (anatomical caudal distal femoral angle), aCdPFA (anatomical caudal proximal femoral angle), aLDFA (anatomical lateral distal femoral angle), aLPFA (anatomical lateral proximal femoral angle), DPA (distal tibial axis/proximal tibial axis angle or diaphyseal proximal tibial angle), FNA (femoral neck angle), FVA (femoral varus angle), ICA (angle of inclination), mCdDFA (mechanical caudal distal femoral angle), mCdPFA (mechanical caudal proximal femoral angle), mCdPTA (mechanical caudal proximal tibial angle), mCrDTA (mechanical cranial distal tibial angle), mLDFA (mechanical lateral distal femoral angle), mLPFA (mechanical lateral proximal femoral angle), mMDTA (mechanical medial distal tibial angle), mMPTA (mechanical medial proximal tibial angle), PA (procurvation angle), Q angle (quadriceps angle), rTTW (relative tibial tuberosity width), TPA (tibial plateau angle), TTA (tibial torsion angle).

**Table 4 animals-11-01804-t004:** Mean ± standard deviation of femoral inclination angle in healthy dogs.

Dog Breeds	Mean (°) ± SD (°) Author [Reference]
Large breeds ^1^	146.2 ± 4.8 Hauptman et al. method A [45] ^Rad^
	129.4 ± 4.9 Hauptman et al. method B [45] ^Rad^
Small breeds ^2^	130 ± 6.5 Olimpo et al. [35] ^Rad^
	129 (117–146) Garnoeva et al. [42] ^Rad^ ^3^
Combination of small and large breeds	148.8 ± 3.7 Montavon et al. [46] ^Rad^
	146.2 ± 5.5 Kara et al. [26] ^CT^
German shepherd	129.9 ± 0.5 Sarierler [27] ^Rad^
	132 ± 5.9 Tomlinson et al. [18] ^Rad^
Labrador retriever	131.6 ± 0.8 Sarierler [27] ^Rad^
	134 ± 5.3 Tomlinson et al. [18] ^Rad^
Pointer	129.8 ± 1.0 Sarierler [27] ^Rad^
Irish setter	128.9 ± 1.5 Sarierler [27] ^Rad^
Anatolian karabash	138.6 ± 1.3 Sarierler [27] ^Rad^
Doberman pinscher	127 ± 1.1 Sarierler [27] ^Rad^
Golden retriever	129.2 ± 2.7 Sarierler [27] ^Rad^
	134 ± 5.2 Tomlinson et al. [18] ^Rad^
Rottweiler	137 ± 5.4 Tomlinson et al. [18] ^Rad^
Pomeranian	136.5 ± 7.1 Soparat et al. [30] ^Rad^
Yorkshire terrier	125.4 ± 4.1 Žilinčík et al. [44] ^Rad^
Toy Poodle	127.7 ± 6.3 Yasukawa et al. [36] ^Rad^
	116.8 ± 6.1 Yasukawa et al. [36] ^CT^
English bulldog	129.1 ± 8 Lusetti et al. [39] ^CT^
English Staffordshire bull terrier	136.7 ± 8.3 Newman and Voss [40] ^CT^
Chihuahua	133.2 ± 7.9 Phetkaew et al. [43] ^Rad^ ^CrCd^
	131.3 ± 3.6 Phetkaew et al. [43] ^Rad^ ^CdCr^
	130.9 ± 4.4 Phetkaew et al. [43] ^CT^

^1^ Combination of different large breeds. ^2^ Combination of different small breeds. ^3^ Values are reported as median (minimum–maximum) in this study. CdCr: caudocranial projection. CrCd: craniocaudal projection. CT: computed tomography. Rad: radiography.

**Table 5 animals-11-01804-t005:** Mean ± standard deviation of femoral inclination angle in dogs with different grades of medial patellar luxation (MPL).

Dog Breeds Author [Reference]	MPL 1 (°)	MPL 2 (°)	MPL 3 (°)	MPL 4 (°)
Small to medium breed				
Mortari et al. [28] ^Rad^	131.2 ± 5.3	130.4 ± 9.5	133.8 ± 1	136.7 ± 4.3
Pomeranian				
Soparat et al. [30] ^Rad^	136.8 ± 6 ^1^		139 ± 9	n/a
Small breeds ^2^				
Olimpo et al. [35] ^Rad^	127.2 ± 3.3	125.3 ± 4.7	130.4 ± 6.2	130 ± 3.5
Garnoeva et al. [42] ^Rad 3^	130 (113–148)	132 (119–168)	138 (110–150)	n/a
Toy Poodle		124.6 ± 7.1 ^Rad^		125.0 ± 6.1 ^Rad^
Yasukawa et al. [36] ^Rad and CT^	n/a	118.0 ± 6.8 ^CT^	n/a	118.3 ± 9.3 ^CT^
English bulldog				
Lusetti et al. [39] ^CT^	124.5 ± 8.3 ^4^			
English Staffordshire bull terrier				
Newman and Voss [40] ^CT^	135.3 ± 7.1 ^4^			
Different or mixed breeds	131.2	132.6	136.4	134.6
Perry et al. [41] ^Rad^	(127.3–135)	(120.9–157.8)	(116.5–163.2)	(126.5–149.5)
Chihuahua Phetkaew et al. [43] ^Rad and CT^	134.1 ± 3.5 ^Rad^ ^CrCd^	132.5 ± 4.4 ^Rad^ ^CrCd^	135.2 ± 8.4 ^Rad^ ^CrCd^	141.7 ± 7.6 ^Rad^ ^CrCd^
	134.0 ± 5.8 ^Rad^ ^CdCr^	134.7 ± 6.4 ^Rad^ ^CdCr^	132.8 ± 7.9 ^Rad^ ^CdCr^	137.5 ± 8.6 ^Rad^ ^CdCr^
	134.6 ± 7.0 ^CT^	133.4 ± 5.2 ^CT^	133.7 ± 4.9 ^CT^	129.5 ± 6.9 ^CT^
Yorkshire terrier				
Žilinčík et al. [44] ^Rad^	125.1 ± 3.7	123.8 ± 7	126.5 ± 4.1	127 ± 4.2

^1^ MPL grades 1 to 2. ^2^ Combination of different small breeds. ^3^ Results are reported as median (minimum–maximum). ^4^ Mean value of four different MPL grades. CdCr: caudocranial projection. CrCd: craniocaudal projection. CT: computed tomography.

**Table 6 animals-11-01804-t006:** Mean ± standard deviation of anatomical lateral proximal femoral angle and mechanical lateral proximal femoral angle in healthy dogs.

Dog Breeds	aLPFA (°) Author [Reference]	mLPFA (°) Author [Reference]
Labrador retriever	103 ± 6.4 Tomlinson et al. [18] ^Rad^	100 ± 6.0 Tomlinson et al. [18] ^Rad^
Golden retriever	98 ± 5.7 Tomlinson et al. [18] ^Rad^	95 ± 5.2 Tomlinson et al. [18] ^Rad^
German shepherd	101 ± 5.0 Tomlinson et al. [18] ^Rad^	97 ± 4.5 Tomlinson et al. [18] ^Rad^
Rottweiler	96 ± 5.3 Tomlinson et al. [18] ^Rad^	93 ± 4.7 Tomlinson et al. [18] ^Rad^
Small breeds ^1^	114.9 ± 8.6 Olimpo et al. [35] ^Rad^ 110 (94–128) Garnoeva et al. [42] ^2^	105.1 ± 4.6 Olimpo et al. [35] ^Rad^ 107 (90–127) Garnoeva et al. [42] ^2^
Toy Poodle	106.6 ± 8.7 Yasukawa et al. [36] ^Rad^ 119.5 ± 5.7 Yasukawa et al. [36] ^CT^	102.1 ± 8.8 Yasukawa et al. [36] ^Rad^ 113.6 ± 6.1 Yasukawa et al. [36] ^CT^
English bulldog	111.7 ± 6.7 Lusetti et al. [39] ^CT^	111 ± 6.9 Lusetti et al. [39] ^CT^
Chihuahua	113 ± 4.2 Phetkaew et al. [43] ^Rad CrCd ^ 112.7 ± 7.6 Phetkaew et al. [43] ^Rad CdCr ^ 124.2 ± 6.6 Phetkaew et al. [43] ^CT^	109.9 ± 7.9 Phetkaew et al. [43] ^Rad CrCd ^ 108.5 ± 8.2 Phetkaew et al. [43] ^Rad CdCr ^ 120 ± 7.1 Phetkaew et al. [43] ^CT^
Yorkshire terrier	118.6 ± 3.4 Žilinčík et al. [44] ^Rad^	n/a

^1^ Combination of different small breeds. ^2^ Results are reported as median (minimum–maximum). aLPFA: anatomical lateral proximal femoral angle. CdCr: caudocranial projection. CrCd: craniocaudal projection. CT: computed tomography. mLPFA: mechanical lateral proximal femoral angle. n/a: not available. Rad: radiography.

**Table 7 animals-11-01804-t007:** Mean ± standard deviation of the anatomical lateral proximal femoral angle and mechanical lateral proximal femoral angle in dogs with different grades of MPL.

Dog Breeds Author [Reference]	Angle	MPL 1 (°)	MPL 2 (°)	MPL 3 (°)	MPL 4 (°)
Small breeds^1^ Olimpo et al. [35] ^Rad^	aLPFA	114 ± 9.1	109.7 ± 8	110.6 ± 8.2	98.3 ± 0
	mLPFA	107.6 ± 7.7	104.6 ± 7.7	106 ± 7.6	93.6 ± 0.5
Small breeds ^1^ Garnoeva et al. [42] ^Rad 2^	aLPFA	106.5 (99–114)	109(91–129)	111 (93–126)	n/a
	mLPFA	106 (100–116)	108 (71–173)	111 (94–130)	n/a
Toy Poodle Yasukawa et al. [36] ^Rad and CT^	aLPFA	n/a	107.6 ± 6.3 ^Rad^ 118.7 ± 4.4 ^CT^	n/a	96.5 ± 8.4 ^Rad^ 112.7 ± 6.8 ^CT^
	mLPFA	n/a	101.5 ± 7.7 ^Rad^ 113.1 ± 3.9 ^CT^	n/a	93.8 ± 5.5 ^Rad^ 109.7 ± 6.4 ^CT^
English Bulldog Lusetti et al. [39] ^CT^	aLPFA	112.2 ± 9.3 ^3^			
	mLPFA	108.1 ± 7.7 ^3^			
Chihuahua Phetkaew et al. [43] ^Rad and CT^	aLPFA	111.2 ± 6.2 ^Rad CrCd^ 108.8 ± 5.4 ^Rad CdCr^ 120.5 ± 4.4 ^CT^	115.9 ± 7.7 ^Rad CrCd^ 110.3 ± 8.6 ^Rad CdCr^ 122.4 ± 7.3 ^CT^	113.8 ± 8.3 ^Rad CrCd^ 110.5 ± 9.7 ^Rad CdCr^ 122.5 ± 7.1 ^CT^	108.5 ± 11.7 ^Rad CrCd^ 103.8 ± 16.4 ^Rad CdCr^ 125.2 ± 7.5 ^CT^
	mLPFA	109 ± 7.3 ^Rad CrCd^ 105.2 ± 5.9 ^Rad CdCr^ 117 ± 5.8 ^CT^	112.6 ± 8.3 ^Rad CrCd^ 107.3 ± 8.6 ^Rad CdCr^ 120.5 ± 8.6 ^CT^	113.4 ± 8.1 ^Rad CrCd^ 107.8 ± 9.4 ^Rad CdCr^ 118.9 ± 8.4 ^CT^	109.6 ± 9.1 ^Rad CrCd^ 104.2 ± 11.4 ^Rad CdCr^ 122.8 ± 7.1 ^CT^
Yorkshire Terrier Žilinčík et al. [44] ^Rad^	aLPFA	120.5 ± 1.9	118.2 ± 6.5	119.6 ± 3.6	94.7 ± 5
	mLPFA	n/a	n/a	n/a	n/a

^1^ Combination of various small breeds. ^2^ The results are reported as median (minimum–maximum) in this study. ^3^ Mean value of all MPL grades. aLPFA: anatomical lateral proximal femoral angle^.^ CdCr: caudocranial projection^.^ CrCd: craniocaudal projection^.^ CT: computed tomography. mLPFA: mechanical lateral proximal femoral angle^.^ MPL: medial patellar luxation^.^ n/a: not available^.^ Rad: radiography.

**Table 8 animals-11-01804-t008:** Mean ± standard deviation of anatomical lateral distal femoral angle, mechanical lateral distal femoral angle, and femoral varus angle in healthy dogs.

Dog Breeds	aLDFA° Author [Reference]	mLDFA° Author [Reference]	FVA° Author [Reference]
Small breeds ^1^	95 ± 3.5 Olimpo et al. [35] ^Rad^ 96 (75–114) Garnoeva et al. [42] ^Rad 2^	103.1 ± 3.4 Olimpo et al. [35] ^Rad^ 100 (84–116) Garnoeva et al. [42] ^Rad 2^	n/a 5.5 (3–23) Garnoeva et al. [42] ^Rad 2^
Medium to large breeds ^3^	n/a	n/a	9.4 ± 2.3 Dudley et al. [16] ^Rad^ 8.8 ± 3.3 Dudley et al. [16] ^CT ^ 7.4 ± 3.9 Dudley et al. [16] ^Ana^
Different breeds	93.3 ± 3.2 Kara et al. [26] ^CT^	n/a	n/a
Labrador retriever	97 ± 3.2 Tomlinson et al. [18] ^Rad^	100 ± 2.6 Tomlinson et al. [18] ^Rad^	n/a
Golden retriever	97 ± 2.8 Tomlinson et al. [18] ^Rad^	100 ± 2.3 Tomlinson et al. [18] ^Rad^	n/a
German shepherd	94 ± 3.3 Tomlinson et al. [18] ^Rad^	97 ± 3.1 Tomlinson et al. [18] ^Rad^	n/a
Rottweiler	98 ± 3.5 Tomlinson et al. [18] ^Rad^	100 ± 2.7 Tomlinson et al. [18] ^Rad^	n/a
Pomeranian	95.2 ± 3.5 Soparat et al. [30] ^Rad^	99.5 ± 4 Soparat et al. [30] ^Rad^	5.8 ± 3.2 Soparat et al. [30] ^Rad^
Toy Poodle	94.4 ± 4.1 Yasukawa et al. [36] ^Rad^ 90.3 ± 2.8 Yasukawa et al. [36] ^CT^	99.1 ± 3.1 Yasukawa et al. [36] ^Rad^ 96.2 ± 2.5 Yasukawa et al. [36] ^CT^	4.4 ± 4.1 Yasukawa et al. [36] ^Rad^ 0.3 ± 2.8 Yasukawa et al. [36] ^CT^
English bulldog	92.3 ± 4.7 Lusetti et al. [39] ^CT^	101.6 ± 2.7 Lusetti et al. [39] ^CT^	n/a
English Staffordshire bull terrier	96.2 ± 4.1 Newman and Voss [40] ^CT^	n/a	n/a
Chihuahua	101.2 ± 4.8 Phetkaew et al. [43] ^Rad^ ^CrCd^ 97.1 ± 3.8 Phetkaew et al. [43] ^Rad CdCr^ 95.7 ± 3.6 Phetkaew et al. [43] ^CT^	102.6 ± 3.1 Phetkaew et al. [43] ^Rad^ ^CrCd^ 101.6 ± 3.2 Phetkaew et al. [43] ^Rad CdCr^ 99.9 ± 3.6 Phetkaew et al. [43] ^CT^	n/a
Yorkshire terrier	95.6 ± 2.1 Žilinčík et al. [44] ^Rad^	n/a	5.6 ± 2.1 Žilinčík et al. [44] ^Rad^

^1^ Combination of various small breeds. ^2^ Results are reported as median (minimum–maximum) in these studies. ^3^ Combination of different breeds. Ana: anatomical specimen. aLDFA: anatomical lateral distal femoral angle. CdCr: caudocranial projection. CrCd: craniocaudal projection. CT: computed tomography. FVA: femoral varus angle. mLDFA: mechanical lateral distal femoral angle. n/a: not available. Rad: radiography.

**Table 9 animals-11-01804-t009:** Mean ± standard deviation of anatomical lateral distal femoral angle, mechanical lateral distal femoral angle, and femoral varus angle in dogs with different grades of MPL.

Dog Breeds Author [Reference]	Angle	MPL 1 (°)	MPL 2 (°)	MPL 3 (°)	MPL 4 (°)
Pomeranian Soparat et al. [30] ^Rad^	aLDFA	98.9 ± 3.9		103.2 ± 5.9	n/a
	mLDFA	101.6 ± 3.1		104.5 ± 4.4	n/a
	FVA	9.4 ± 3.7		13.1 ± 5.5	n/a
Small Breeds Olimpo et al. [35] ^Rad^	aLDFA	100 ± 4	95.6 ± 6	98.6 ± 7	107 ± 14.6
	mLDFA	103.3 ± 3	99.8 ± 4.5	103.5 ± 6.2	105 ± 5.6
	FVA	n/a	n/a	n/a	n/a
Toy Poodle Yasukawa et al. [36] ^Rad and CT^	aLDFA	n/a	94.3 ± 4.8 ^Rad^ 89.5 ± 3.8 ^CT^	n/a	110.5 ± 8.5 ^Rad^ 108.1 ± 8.0 ^CT^
	mLDFA	n/a	99.3 ± 3.9 ^Rad^ 95.0 ± 3.6 ^CT^	n/a	113.3 ± 5.3 ^Rad^ 111.1 ± 6.9 ^CT^
	FVA	n/a	4.3 ± 4.8 ^Rad^ 0.6 ± 3.8 ^CT^	n/a	20.5 ± 8.5 ^Rad^ 18.1 ± 8.0 ^CT^
English bulldog Lusetti et al. [39] ^CT^	aLDFA	n/a	100 ± 8.4		n/a
	mLDFA	n/a	103.2 ± 4.4		n/a
	FVA	n/a	n/a		n/a
English Staffordshire bull terrier Newman and Voss [40] ^CT^	aLDFA	98.6 ± 3.2 *			
	mLDFA	n/a			
	FVA	n/a			
Small breeds Garnoeva et al. [42] ^Rad^ **	aLDFA	102 (92–118)	106 (86–129)	109 (84–125)	n/a
	mLDFA	102 (96–114)	105 (85–127)	109 (92–119)	n/a
	FVA	13 (7–17)	17 (2–36)	18 (3–27)	n/a
Chihuahua Phetkaew et al. [43] ^Rad and CT^	aLDFA	99.8 ± 4.8 ^Rad CrCd^ 99.4 ± 5.0 ^Rad CdCr^ 97 ± 4.2 ^CT^	100.7 ± 3.0 ^Rad CrCd^ 100.8 ± 3.5 ^Rad CdCr^ 97.6 ± 3.6 ^CT^	102.7 ± 3.1 ^Rad CrCd^ 102.1 ± 5.1 ^Rad CdCr^ 98.7 ± 4.2 ^CT^	114.6 ± 11.5 ^Rad CrCd^ 112.1 ± 13.3 ^Rad CdCr^ 109.2 ± 9.7 ^CT^
	mLDFA	101 ± 6.3 ^Rad CrCd^ 102.7 ± 3.3 ^Rad CdCr^ 100 ± 2 ^CT^	103.2 ± 2 ^Rad CrCd^ 103.4 ± 2.5 ^Rad CdCr^ 101.3 ± 2.5 ^CT^	104.6 ± 2.2 ^Rad CrCd^ 104.6 ± 3 ^Rad CdCr^ 102.7 ± 3.3 ^CT^	113.5 ± 8 ^Rad CrCd^ 112.1 ± 8.8 ^Rad CdCr^ 111.9 ± 9.3 ^CT^
	FVA	n/a	n/a	n/a	n/a
Yorkshire terrier Žilinčík et al. [44] ^Rad^	aLDFA	96.1 ± 2	97.2 ± 3.4	100.5 ± 2	110.2 ± 6.6
	mLDFA	n/a	n/a	n/a	n/a
	FVA	6.1 ± 2	6.9 ± 2.7	10.5 ± 2	20.3 ± 6.6

* Mean and standard deviation for different grades of MPL (MPL 1–4). ** Results are reported as median (minimum–maximum) in these studies. aLDFA: anatomical lateral distal femoral angle. CdCr: caudocranial projection. CrCd: craniocaudal projection. CT: computed tomography. FVA: femoral varus angle. mLDFA: mechanical lateral distal femoral angle. MPL: medial patellar luxation. n/a: not provided. Rad: radiography.

**Table 10 animals-11-01804-t010:** Mean ± standard deviation of the quadriceps angle in healthy dogs and dogs with MPL.

Dog Breeds Author [Reference]	Healthy Dogs	MPL 1 (°)	MPL 2 (°)	MPL 3 (°)	MPL 4 (°)
Combination of different breeds				
Kaiser et al. [48] ^MRI^ Mortari et al. [28] ^Rad^ Pinna and Romagnoli. [25] ^Rad^ * Pinna and Romagnoli. [25] ^Rad^ ** Garnoeva et al. [42] ^Rad^	10.9 ± 5.6 n/a 18.3 (6.1–29.7) *** 8.7 (2.7–14.8) *** 14 (8–28) ***	10.1 ± 8.5 14.9 ± 7 n/a n/a 20.5 (14–30) ***	20.6 ± 10.9 22.1 ± 6.4 n/a n/a 22 (15–39) ***	38.1 ± 8.6 34.4 ± 13.7 n/a n/a 31 (18–46) ***	n/a 34.0 ± 9.4 n/a n/a n/a
German shepherd Łojszczyk-Szczepaniak et al. [22] ^Rad^	17° ± 7.4	n/a	n/a	n/a	n/a

* Dogs with bodyweight below 15 kg. ** Dogs with bodyweight more than 15 kg. *** Results are reported as median (minimum–maximum) in these studies. MPL: medial patellar luxation. MRI: magnetic resonance imaging. n/a: not provided. Q angle: quadriceps angle. Rad: radiography.

**Table 11 animals-11-01804-t011:** Mean ± standard deviation of the anatomical caudal proximal femoral angle, mechanical caudal proximal femoral angle, anatomical caudal distal femoral angle, mechanical caudal distal femoral angle, and procurvatum angle, as recorded in the included literature.

Dog Breeds Author [Reference]	Angles	Healthy Dogs (°)	MPL 1 (°)	MPL 2 (°)	MPL 3 (°)	MPL 4 (°)
Toy Poodle Yasukawa et al. [36]	aCdPFA	157.3 ± 7.7 ^Rad^ 153.3 ± 5.1 ^CT^	n/a	153.3 ± 8.0 ^Rad^ 151.6 ± 6.0 ^CT^	n/a	152.5 ± 11.3 ^Rad^ 151.7 ± 5.6 ^CT^
	mCdPFA	7.5 ± 5.9 ^Rad^ 9.6 ± 5.5 ^CT^	n/a	10.6 ± 7.5 ^Rad^ 11.3 ± 5.9 ^CT^	n/a	13.4 ± 8.8 ^Rad^ 10.4 ± 6.2 ^CT^
	aCdDFA	104.3 ± 2.1 ^Rad^ 102.9 ± 3.2 ^CT^	n/a	104.5 ± 5.6 ^Rad^ 102.6 ± 3.5 ^CT^	n/a	105.6 ± 6.9 ^Rad^ 104.7 ± 5.7 ^CT^
	mCdDFA	107.8 ± 1.9 ^Rad^ 108.4 ± 1.7 ^CT^	n/a	107.0 ± 3.7 ^Rad^ 107.5 ± 2.6 ^CT^	n/a	107.5 ± 1.8 ^Rad^ 107.0 ± 2.7 ^CT^
	PA	12.7 ± 4.1 ^Rad^ 11.2 ± 5.2 ^CT^	n/a	12.7 ± 7.1 ^Rad^ 11.1 ± 5.4 ^CT^	n/a	14.2 ± 7.3 ^Rad^ 15.8 ± 6.9 ^CT^
Chihuahua Phetkaew et al. [43]	aCdPFA	148.5 ± 4.8 ^Rad^ 156.4 ± 5.3 ^CT^	152.9 ± 8.4 ^Rad^ 155.1 ± 7.8 ^CT^	148.0 ± 7.0 ^Rad^ 149.6 ± 4.6 ^CT^	152.6 ± 7.0 ^Rad^ 148.6 ± 6.8 ^CT^	142.6 ± 8.3 ^Rad^ 147.3 ± 6.6 ^CT^
	aCdDFA	103.8 ± 2.6 ^Rad^ 106.2 ± 2.4 ^CT^	101.3 ± 1.9 ^Rad^ 103.6 ± 2.9 ^CT^	100.0 ± 4.5 ^Rad^ 101.8 ± 4.0 ^CT^	100.8 ± 3.3 ^Rad^ 102.7 ± 4.4 ^CT^	102.3 ± 5.5 ^Rad^ 102.6 ± 6.4 ^CT^
	PA	9.1 ± 2.9 ^Rad^ 11.7 ± 3.4 ^CT^	7.3 ± 2.7 ^Rad^ 10.1 ± 3.4 ^CT^	7.0 ± 3.4 ^Rad^ 7.5 ± 2.9 ^CT^	7.2 ± 5.2 ^Rad^ 8.4 ± 6.2 ^CT^	6.9 ± 5.0 ^Rad^ 9.9 ± 5.6 ^CT^
Different breeds Kara et al. [26]	aCdDFA	90.51 ± 6.19 ^CT^	n/a	n/a	n/a	n/a

aCdPFA: anatomical caudal proximal femoral angle. aCdDFA: anatomical caudal distal femoral angle. CT: computed tomography. mCdPFA: mechanical caudal proximal femoral angle. mCdDFA: mechanical caudal distal femoral angle. MPL: medial patellar luxation. n/a: not provided. Rad: radiography.

**Table 12 animals-11-01804-t012:** Mean ± standard deviation of the AA in healthy dogs.

Dog Breeds	AA (°) Authors [Reference]
Combination of different breeds	16 ± 6.4 Dudley et al. [16] ^Rad^ 19.6 ± 7.9 Dudley et al. [16] ^CT^ 18.9 ± 5.4 Dudley et al. [16] ^Ana ^ 26.9 ± 11.5 Kara et al. [26] ^CT^
Small breeds	20.4 ± 4.8 Olimpo et al. [35] ^Rad^
Toy Poodle	19.8 ± 4.6 Yasukawa et al. [36] ^CT^
English bulldog	11.4 ± 6.4 Lusetti et al. [39] ^CT^
English Staffordshire bull terrier	26.0 ± 3.4 Newman and Voss [40] ^CT^
Chihuahua	29.2 ± 6.3 Phetkaew et al. [43] ^CT^
Yorkshire terrier	19.6 ± 2.9 Žilinčík et al. [44] ^Rad^

AA: angle of anteversion. Ana: anatomical specimen. CT: computed tomography. Rad: radiography.

**Table 13 animals-11-01804-t013:** Mean ± standard deviation of the angle of anteversion in dogs with different grades of MPL.

Dog Breeds Authors [Reference]	MPL 1 (°)	MPL 2 (°)	MPL 3 (°)	MPL 4 (°)
Small breeds Olimpo et al. [35] ^Rad^	17.8 ± 3.8	104.6 ± 7.4 *	15.2 ± 8	17 ± 0
Toy Poodle Yasukawa et al. [36] ^CT^	n/a	16.6 ± 4.8	n/a	9.6 ± 5.2
English bulldog Lusetti et al. [39] ^CT^	6.9 ± 12.8 **			
English Staffordshire bull terrier Newman and Voss [40] ^CT^	n/a	21.9 ± 3.7		n/a
Chihuahua Phetkaew et al. [43] ^CT^	25.9 ± 7.8	27.6 ± 6.5	25.8 ± 6.0	21.1 ± 5.6
Yorkshire terrier Žilinčík et al. [44] ^Rad^	19.2 ± 1.9	19.1 ± 2.6	17.0 ± 2.2	9.2 ± 2.8

* The results reported for MPL grade 2 in this study do not match with other grades, and this seems to be due to a misspelling in the original article. ** Mean and standard deviation for different grades of MPL (grade 1–4 MPL). CT: computed tomography. MPL: medial patellar luxation. n/a: not provided. Rad: radiography.

**Table 14 animals-11-01804-t014:** Mean ± standard deviation of the mechanical medial proximal tibial angle and mechanical medial distal tibial angle in healthy dogs.

Dog Breed	mMPTA (°) Author [Reference]	mMDTA (°) Author [Reference]
Small breeds	95.1 ± 3.2 Olimpo et al. [35] ^Rad ^ 90 (78–108) * Garnoeva et al. [42] ^Rad^	98.1 ± 4.4 Olimpo et al. [35] ^Rad ^ 90 (75-99) * Garnoeva et al. [42] ^Rad^
English bulldog	92.0 ± 4.3 Lusetti et al. [39] ^CT^	91.3 ± 3.0 Lusetti et al. [39] ^CT^
Chihuahuas	94.0 ± 1.0 Phetkaew et al. [43] ^Rad CrCd^ 99.1 ± 2.2 Phetkaew et al. [43] ^Rad CdCr^ 96.3 ± 4.1 Phetkaew et al. [43] ^CT^	97.2 ± 3.7 Phetkaew et al. [43] ^Rad CrCd^ 93.4 ± 1.1 Phetkaew et al. [43] ^Rad CdCr^ 94.3 ± 7.8 Phetkaew et al. [43] ^CT^

* Results are reported as median (minimum–maximum) in this study. CdCr: caudocranial projection. CrCd: craniocaudal projection. CT: computed tomography. mMPTA: mechanical medial proximal tibial angle. mMDTA: mechanical medial distal tibial angle. n/a: not provided. Rad: radiography.

**Table 15 animals-11-01804-t015:** Mean ± standard deviation of the mechanical medial proximal tibial angle and mechanical medial distal tibial angle in dogs with MPL and CrCL disease.

Dog Breed	Health Condition	mMPTA (°) Author [Reference]	mMDTA (°) Author [Reference]
Labrador retriever	CrCL diseases	93.4 ± 1.8 Dismukes et al. [19] ^Rad^	96.3 ± 2.5 Dismukes et al. [19] ^Rad^
Medium to large breed dogs	CrCL diseases	93.3 ± 1.8 Dismukes et al. [19] ^Rad^	96.0 ± 2.70 Dismukes et al. [19] ^Rad^
		92.2 ± 1.8 Dismukes et al. [20] ^Rad^	95.9 ± 2.2 Dismukes et al. [20] ^Rad^
Combination of different breeds	Bilateral CrCL	93.3 ± 1.8 Fuller et al. [32] ^Rad^	95.8 ± 1.9 Fuller et al. [32] ^Rad^
	Unilateral CrCL ^§^	92.6 ± 2.2 Fuller et al. [32] ^Rad^	95.6 ± 1.9 Fuller et al. [32] ^Rad^
	Unilateral CrCL ^+^	93.1 ± 2.6 Fuller et al. [32] ^Rad^	94.9 ± 2.0 Fuller et al. [32] ^Rad^
Small breeds	MPL 1	95.1 ± 2.5 Olimpo et al. [35] ^Rad^	96 ± 3.3 Olimpo et al. [35] ^Rad^
		90 (81–103) Garnoeva et al. [42] ^Rad ¥^	96 (83–106) Garnoeva et al. [42] ^Rad ¥^
	MPL 2	94.8 ± 2 Olimpo et al. [35] ^Rad^	97.2 ± 3.9 Olimpo et al. [35] ^Rad^
		92 (85–107) Garnoeva et al. [42] ^Rad ¥^	90 (75–103) Garnoeva et al. [42] ^Rad ¥^
	MPL 3	97.1 ± 4.7 Olimpo et al. [35] ^Rad^	97.1 ± 3.8 Olimpo et al. [35] ^Rad^
		97 (87–110) Garnoeva et al. [42] ^Rad ¥^	90 (79–100) Garnoeva et al. [42] ^Rad ¥^
	MPL 4	110.8 ± 12.5 Olimpo et al. [35] ^Rad^	96.2 ± 2.7 Olimpo et al. [35] ^Rad^
Toy Poodle	MPL 2	96.9 ± 3.5 Yasukawa et al. [36] ^Rad^	94.2 ± 4.4 Yasukawa et al. [36] ^Rad^
		94.7 ± 1.7 Yasukawa et al. [36] ^CT^	95.2 ± 2.4 Yasukawa et al. [36] ^CT^
	MPL 4	94.5 ± 4.4 Yasukawa et al. [36] ^CT^	98.5 ± 4.1 Yasukawa et al. [36] ^CT^
English bulldogs	MPL (1–4) ^ǂ^	93.2 ± 4.3 Lusetti et al. [39] ^CT^	93.0 ± 3.0 Lusetti et al. [39] ^CT^
Chihuahua	MPL 1	96.6 ± 3.1 Phetkaew et al. [43] ^Rad CrCd^	92.3 ± 4.3 Phetkaew et al. [43] ^Rad CrCd^
		96.9 ± 3.1 Phetkaew et al. [43] ^Rad CdCr^	94.8 ± 3.5 Phetkaew et al. [43] ^Rad CdCr^
		95.8 ± 3.0 Phetkaew et al. [43] ^CT^	92.0 ± 4.7 Phetkaew et al. [43] ^CT^
	MPL 2	94.7 ± 3.3 Phetkaew et al. [43] ^Rad CrCd^	93.6 ± 3.9 Phetkaew et al. [43] ^Rad CrCd^
		97.1 ± 3.3 Phetkaew et al. [43] ^Rad CdCr^	93.3 ± 2.4 Phetkaew et al. [43] ^Rad CdCr^
		96.7 ± 3.3 Phetkaew et al. [43] ^CT^	92.6 ± 4.4 Phetkaew et al. [43] ^CT^
	MPL 3	96.2 ± 2.3 Phetkaew et al. [43] ^Rad CrCd^	92.1 ± 2.7 Phetkaew et al. [43] ^Rad CrCd^
		98.4 ± 2.7 Phetkaew et al. [43] ^Rad CdCr^	95.0 ± 2.4 Phetkaew et al. [43] ^Rad CdCr^
		96.7 ± 3.3 Phetkaew et al. [43] ^CT^	91.9 ±2.6 Phetkaew et al. [43] ^CT^
	MPL 4	99.6 ± 7.1 Phetkaew et al. [43] ^Rad CrCd^	100.3 ± 6.2 Phetkaew et al. [43] ^Rad CrCd^
		103.1 ± 7.2 Phetkaew et al. [43] ^Rad CdCr^	97.3 ± 4.2 Phetkaew et al. [43] ^Rad CdCr^
		102.2 ± 8.5 Phetkaew et al. [43] ^CT^	94.4 ± 3.9 Phetkaew et al. [43] ^CT^

^§^ Unilateral cranial cruciate ligament rupture with subsequent contralateral rupture. ^+^ Unilateral cranial cruciate ligament rupture without subsequent contralateral rupture. ^¥^ Results are reported as median (minimum–maximum) in this study. ^‡^ Mean and standard deviation for different grades of MPL (grade 1–4 MPL). CdCr: caudocranial projection. CrCd: craniocaudal projection. CrCL: cranial cruciate ligament. CT: computed tomography. MPL: medial patellar luxation. mMPTA: mechanical medial proximal tibial angle. mMDTA: mechanical medial distal tibial angle. Rad: radiography.

**Table 16 animals-11-01804-t016:** Mean ± standard deviation of the tibial alignments in the sagittal plane.

Dog Breed	Study [Reference]	TPA (°)	DPA (°)	mCdPTA (°)	mCrDTA (°)	Z Angle (°)	rTTW (°)
Large breeds	Osmond et al. [17] ^Rad^	n/a	Healthy: 4.1 ± 2.2 CrCL: 6.0 ± 3.3	n/a	n/a	n/a	n/a
	Dismukes et al. [21] ^Rad^	n/a	n/a	CrCL: 63 ± 3.9	CrCL: 81.5 ± 4.1	n/a	n/a
	Fuller et al. [32] ^Rad^	Bi-CrCL ^1^: 26.4 ± 3.8 Uni-CrCL ^2^: 27.0 ± 3.9 Uni-wo ^3^: 28 ± 3.6	n/a	Bi-CrCL: 63.6 ± 3.8 Uni-CrCL: 63.0 ± 3.9 Uni-wo: 62.0 ± 3.6	Bi-CrCL: 80.5 ± 3.2 Uni-CrCL: 79.7 ± 2.8 Uni-wo: 80.8 ± 3.4	n/a	n/a
	Aertsens et al. [33] ^Rad^	CrCL: 24.9 ± 3.9	n/a	n/a	n/a	CrCL: 64.0 ± 4.7	CrCL: 0.8 ± 0.1
	Su et al. [34] ^Rad^	Healthy: 26.1 ± 0.8	n/a	n/a	n/a	n/a	n/a
	Guénégo et al. [37] ^Rad 4^	Healthy: 24.0 (10.40–34.00) CrCL: 27.5 (20.0–42.0)	n/a	n/a	n/a	Healthy: 63.0 (54.0–72.50) CrCL: 64.30 9(52.0–83.2)	Healthy: 0.84 (0.69–1.26) CrCL: 0.73 (0.55–0.98)
Labrador retriever	Dismukes et al. [21] ^Rad^	n/a	n/a	CrCL: 63.8 ± 3.7	CrCL: 81.7 ± 4.2	n/a	n/a
	Ragetly et al. [29] ^Rad^	Healthy: 25.2 ± 2.1 Predisposed: 28.4 ± 2.0	n/a	n/a	n/a	n/a	n/a
	Vedrine et al. [31] ^Rad^	Healthy: 25 ± 3	Healthy: 4.5 ± 2.3	n/a	n/a	n/a	n/a
Yorkshire terrier	Vedrine et al. [31] ^Rad^	Healthy: 30 ± 4	Healthy: 10.8 ± 4.3	n/a	n/a	n/a	n/a
Medium to large breeds	Sabanci and Ocal [23] ^Rad^	Healthy (medial): 24.0 ± 3.2 Healthy (lateral): 25.5 ± 3.8	n/a	n/a	n/a	n/a	n/a
Small breeds	Aertsens et al. [33] ^Rad^	CrCL: 30.1 ± 5.3	n/a	n/a	n/a	CrCL: 70.0 ± 5.6	CrCL: 0.8 ± 0.1
	Su et al. [34] ^Rad^	Healthy: 29.2 ± 0.8	n/a	n/a	n/a	n/a	n/a
	Witte [24] ^Rad^	Healthy: 32 ± 6.2	Healthy: 10.2 ± 7.3	n/a	n/a	n/a	n/a
	Olimpo et al. [35] ^Rad^	Healthy: 24.4 ± 3.0 MPL1: 24.6 ± 3.9 MPL2: 23 ± 3.7 MPL3: 23.2 ± 5 MPL4: 16.6 ± 10.4	n/a	Healthy: 65 ± 3.02 MPL1: 74 ± 4.3 MPL2: 72.5 ± 4.3 MPL3: 74 ± 5.5 MPL4: 69.6 ± 5.2	Healthy: 86.3 ± 1.5 MPL1: 84.6 ± 2.7 MPL2: 82.6 ± 1.5 MPL3: 86.8 ± 2.1 MPL4: 87 ± 0	n/a	n/a
	Janovec et al. [38] ^Rad^	Healthy: 29.2 ± 7.3 CrCL: 32.0 ± 5.7	n/a	n/a	n/a	n/a	n/a
	Garnoeva et al. [42] ^Rad 4^	n/a	n/a	Healthy: 63 (54–84) MPL1: 60 (54–84) MPL2: 61 (29–74) MPL3: 64 (51–77)	Healthy: 91 (70–101) MPL1: 89 (70–104) MPL2: 84 (68–108) MPL3: 90(76–104)	n/a	n/a
Toy Poodle	Yasukawa et al. [36]	Healthy ^Rad^: 27.6 ± 4.7 Healthy ^CT^: 21.3 ± 3.3 MPL2 ^Rad^: 28.4 ± 5.3 MPL2 ^CT^: 21.2 ± 3.4 MPL4 ^CT^: 22.7 ± 4.2	n/a	n/a	Healthy ^Rad^: 91.0 ± 4.6 Healthy ^CT^: 98.5 ± 3.8 MPL2 ^Rad^: 88.8 ± 2.0 MPL2 ^CT^: 99.2 ± 3.1 MPL4 ^CT^: 98.6 ± 6.4	Healthy ^Rad^: 63.8 ± 5.2 Healthy ^CT^: 65.7 ± 4.6 MPL2 ^Rad^: 64.5 ± 3.9 MPL2 ^CT^: 66.2 ± 3.8 MPL4 ^CT^: 67.2 ± 5.8	Healthy ^Rad^: 0.9 ± 0.1 Healthy ^CT^: 0.7 ± 0.1 MPL2 ^Rad^: 64.5 ± 3.9 MPL2 ^CT^: 66.2 ± 3.8 MPL4 ^CT^: 67.2 ± 5.8
English bulldog	Lusetti et al. [39] ^CT^	n/a	n/a	Healthy: 63.2 ± 6.1 MPL (1-4)^5^: 66.0 ± 10.4	n/a	n/a	n/a
Chihuahua	Phetkaew et al. [43]	n/a	n/a	Healthy ^Rad^: 63.1 ± 1.2 Healthy ^CT^: 65.3 ± 2.6 MPL1 ^Rad^: 63.5 ± 4.1 MPL1 ^CT^: 63.9 ± 4.6 MPL2 ^Rad^: 64.1 ± 2.3 MPL2 ^CT^: 62.3 ± 3.5 MPL3 ^Rad^: 63.9 ± 4.1 MPL3 ^CT^: 62.1 ± 5.2 MPL4 ^Rad^: 65.1 ± 3.3 MPL4 ^CT^: 59.9 ± 4.8	Healthy ^Rad^: 92.0 ± 2.4 Healthy ^CT^: 94.9 ± 3.1 MPL1 ^Rad^: 92.2 ± 4.0 MPL1 ^CT^: 91.7 ± 5.1 MPL2 ^Rad^: 88.0 ± 2.3 MPL2 ^CT^: 91.9 ± 4.3 MPL3 ^Rad^: 91.8 ± 4.1 MPL3 ^CT^: 91.4 ± 5.4 MPL4 ^Rad^: 88.3 ± 4.4 MPL4 ^CT^: 96.4 ± 3.7	n/a	n/a

^1^ Bilateral CrCL. ^2^ Unilateral CrCL with subsequent contralateral rupture. ^3^ Unilateral CrCL without subsequent contralateral rupture. ^4^ Values are expressed as median (minimum–maximum) in these studies. ^5^ Mean and standard deviation for different grades of MPL (grade 1–4 MPL). CrCL: cranial cruciate ligament. CT: computed tomography. DPA: diaphyseal proximal tibial angle. mCdPTA: mechanical caudal proximal tibial angle. mCrDTA: mechanical cranial distal tibial angle. MPL: medial patellar luxation. n/a: not available. Rad: radiograph. rTTW: relative tibial tuberosity width. TPA: tibial plateau angle.

## Data Availability

No new data were created or analyzed in this study. Data sharing is not applicable to this article.

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
