# Peer review of "Evaluation of the Femoral and Tibial Alignments in Dogs: A Systematic Review"

_animals, 2021, doi:10.3390/ani11061804_

Round 1
Reviewer 1 Report
Manuscript "Evaluation of the Femoral and Tibial Alignments in Dogs: A Systematic Review : is very well prepared and presented review article. Authors analized world , scientific litarature very completely and deeply in the area of femoral angles, tibial angles, limb alignment,bone deformity. Besides various methods of measurement, orthopedic diseases , and breeds of dogs with above mentioned problems were presented too. In my opinin this article took femoral and tibial alignments problem very holistically up. One editorial remark : in references , posioton 22,(line 949) have to be write in normal type. To authors consider : to cite publication by : Adam Przeworski and others , in Medycyna Weterynaryjna, title: Tibial plateau angle: methods of measurement , value, applicability.
Author Response
Author's Reply to the Review Report (Reviewer 1)
Dear Reviewer,
Thank you for your comments on our manuscript. The recommendations made (minor revision) have added clarification and quality to our study.
We feel that the revised manuscript has addressed the concerns raised during the review process. We hope that the revised manuscript is suitable for publication and look forward to hearing from you.
Reviewer comment:
One editorial remark: in references, position 22, (line 949) have to be write in normal type. To authors consider to cite publication by: Adam Przeworski and others, in Medycyna Weterynaryjna, title: Tibial plateau angle: methods of measurement, value, applicability.
Response:
- As suggested, this correction has been made (Reference 22, Line 949).
- According to the eligibility criteria of the study, the full text of all included articles had to be published in English (Line 115), and articles in other languages were excluded; therefore, despite the scientific value of the recommended article, we did not include it in our final list.
Yours Sincerely
Masoud Aghapour

Reviewer 2 Report
The manuscript is an exhaustive systematic review that aims to evaluate, from literature, femural and tibial alignment to report, for each alignment, the reference values for healthy dogs and the differences between healthy and sick dogs.
The paper appears very clear and the authors have chosen a scientific review tool (PRISMA statement) which increases the scientific value of the work.
The topic is very broad but is well analyzed by the authors in many aspects.
I recommend reviewing the discussions as they appear a bit repetitive (the main goal is repeated at least three times) which reduces the readability of the manuscript.
Author Response
Author's Reply to the Review Report (Reviewer 2)
Dear Reviewer,
Thank you for your kind comments and suggestions for our manuscript. Your recommendations made (minor revision) have added clarification and quality to our study. We feel that the revised manuscript has addressed the concerns raised during the review process. We hope that the revised manuscript is suitable for publication and look forward to hearing from you.
Reviewer comment:
I recommend reviewing the discussions, as they appear a bit repetitive (the main goal is repeated at least three times) which reduces the readability of the manuscript.
Response:
As suggested, the discussion part has been revised and the duplicate sentences have been removed as below:
- Line 740 - 743: These sentences have been deleted as they have been mentioned previously in results (line 138 – 149 and line 190 - 196).
- Line 746 - 748: These sentences have been deleted as they have been mentioned previously in results (line 138 - 149).
- Line 748-754: These sentences have been deleted as they have been mentioned at the end of the discussion in study limitations (Line 858).
Yours Sincerely
Masoud Aghapour

Reviewer 3 Report
I have not comments, the paper is well organized and the results clear.
Author Response
Author's Reply to the Review Report (Reviewer 3)
Dear Reviewer,
Thank you for your kind comments and the valuable time that you spent on our manuscript. The manuscript has been revised (minor revision) based on the opinion of the other Reviewers. We feel that the revised manuscript has addressed the concerns raised during the review process. We hope that the revised manuscript is suitable for publication and look forward to hearing from you.
Yours Sincerely
Masoud Aghapour
